# Biodiverse food plants in the semiarid region of Brazil have unknown potential: A systematic review

**Michelle Cristine Medeiros Jacob** [1]\*, **Maria Fernanda Araújo de Medeiros**[1], **Ulysses Paulino Albuquerque**[2]

**1** Laboratório Horta Comunitária Nutrir, Nutrition Department, Universidade Federal do Rio Grande do Norte, Natal, Rio Grande do Norte, Brazil, **2** Botany Department, Laboratório de Ecologia e Evolução de Sistemas Socioecológicos, Universidade Federal de Pernambuco, Recife, Pernambuco, Brazil

\* michellejacob@ufrn.edu.br

**Data Availability Statement:** All relevant data are within the manuscript and its supporting information files.

## Abstract

Food biodiversity presents one of the most significant opportunities to enhance food and nutrition security today. The lack of data on many plants, however, limits our understanding of their potential and the possibility of building a research agenda focused on them. Our objective with this systematic review was to identify biodiverse food plants occurring in the Caatinga biome, Brazil, strategic for the promotion of food and nutrition security. We selected studies from the following databases: Web of Science, Medline/PubMed (via the National Library of Medicine), Scopus and Embrapa Agricultural Research Databases (BDPA). Eligible were original articles, published since 2008, studying food plants occurring in the Caatinga. We assessed the methodological quality of the studies we selected. We reviewed a total of fifteen studies in which 65 plants that met our inclusion criteria were mentioned. Of this amount, 17 species, including varieties, subspecies, and different parts of plants, had data on chemical composition, in addition to being mentioned as food consumed by rural communities in observational ethnobotanical studies. From the energy and protein data associated with these plants, we produced a ranking of strategic species. The plants with values higher than the average of the set were: *Dioclea grandiflora* Mart. ex Benth (mucunã), *Hymenaea courbaril* L. (jatobá), *Syagrus cearensis* Noblick (coco-catolé), *Libidibia ferrea* (Mart. ex Tul.) L.P.Queiroz (jucá), *Sideroxylon obtusifolium* (Roem. & Schult.) T. D.Penn. (quixabeira). We suggest that the scientific community concentrates research efforts on tree legumes, due to their resilience and physiological, nutritional, and culinary qualities.

## Introduction

The scientific community pinpoints the reform of food systems as one of the main actions to face the Global Syndemic of obesity, undernutrition, and climate change [1–4]. This reform involves promoting sustainable diets, which connect the challenges of food and nutrition

**Funding:** This study was funded by the Universidade Federal do Rio Grande do Norte through a Scientific Initiation research scholarship to MFAM (UFRN call 01/2018), and by the National Council for Scientific and Technological Development through a postdoctoral scholarship to MCMJ (150654/2019-7) and research productivity scholarship granted to UPA (302380/2011-6). The Brazilian Coordenação de Aperfeiçoamento de Pessoal de Nível financed the fee to publish this article (Finance Code 001). Funders had no role in the study design, data collection and analysis, decision to publish or prepare the manuscript.

**Competing interests:** The authors have declared that no competing interests exist.

security (FNS) and biodiversity conservation, expressed in objectives 2 and 15 in the United Nations 2030 agenda [5].

There is no doubt that the approach to sustainable diets is associated with the need to map the available food biodiversity [6]. There are a variety of publications that already present data of this nature. They are observational ethnobotanical studies, experimental research on the chemical composition of food, ethnographic analyses, which are dispersed, separated in different areas of knowledge: health, environmental, and agrarian sciences, as well as humanities, among others. With this systematic review (SR), we seek to connect these data to provide the state of available and known food biodiversity in one of the Brazilian ecosystems most threatened by degradation processes associated with climate change, the Caatinga (dry seasonal forest). Considering that disciplinary barriers limit our perception of the problem of FNS, we intend to lay the groundwork for a research agenda that includes the multiple disciplinary perspectives involved in the analysis of FNS.

Brazil has an estimated flora of 46,833 species, including algae, angiosperms, bryophytes, fungi, gymnosperms, ferns and lycophytes [7]. A total of 6,053 of these species occur in the Caatinga, one of the six Brazilian biomes, distributed over an area of 844,453 $km^2$, which corresponds to almost 10% of the national territory. The Caatinga, where about 27 million people live, is a region with successive periods of drought, hot weather, and xerophytic vegetation [8]. We justify the choice of this biome as part of this review in two ways. First, the accelerated process of degradation via anthropic action highlights the urgency of finding strategies to protect its species diversity. Second, the fact that the Caatinga covers the region of Brazil, the Northeast, with the second highest prevalence of severe food insecurity (hunger proxy) in the country [9], also is a rationale of our choice.

Evidence indicates that food biodiversity is one of the factors positively correlated with the quality of diets. In a study to evaluate the nutritional adequacy and dietary biodiversity of the diets of women and children in rural areas of Benin, Cameroon, Democratic Republic of Congo, Ecuador, Kenya, Sri Lanka, and Vietnam, Lachat *et al.* [10] observed a positive association between the species richness of food consumed and the quality of the diet, both in dry and rainy seasons. They presented relevant data for policymakers in developing countries since the global biodiversity hotspots coincide with areas of low income, poverty, and undernutrition [11]. Other references emphasize the crucial role that native plants play in supplementing essential micronutrients, providing a safety net during periods of scarcity [12]. Besides, there is well-established evidence that links food diversity to the adequacy of energy, micronutrients, and child growth [13].

On the other hand, we must consider that an environment rich in biodiversity does not necessarily contribute to better quality diets. This is shown by the food consumption assessment study carried out in the Democratic Republic of Congo by Termote *et al.* [14]. The authors found that in this region of high biodiversity and with the population experiencing severe food insecurity, the consumption of local plants was insufficient, limiting the adequacy of diets. The authors listed the lack of information about these plants as one of the probable reasons for their low consumption. Undoubtedly, one of the challenges involved in promoting sustainable diets is the scarcity of data on availability, consumption, and nutritional composition of these kinds of plants, which we will call biodiverse food plants here [6,15]. We consider biodiverse food plants (BFP) as plants of extensive use (e.g., beans, rice, corn) and unconventional food plants (UFP), usually native, often neglected, and of culturally-limited use. For UFP, we can also consider native and heirloom varieties of conventional foods grown locally. In conventional dietary surveys, the consumption of BFP is often not analyzed, which is a cause and a consequence of the absence of these species in food composition tables. It is a cause because it is unproductive to collect data that will not be adequately analyzed. It is a

consequence because it is not productive to conduct food composition studies on plants that, theoretically, are not consumed. The lack of data of this nature is more significant in the case of the UFP [16].

Therefore, with this SR, our objective is to answer the following question: *Which food plants occurring in the Caatinga biome are strategic for promoting food and nutrition security*? For this, we listed and characterized food plants occurring in the Caatinga mentioned in the reviewed studies, and then we selected strategic plants to promote FNS. To date, there is no SR study on food plants in the Caatinga.

## Method

This SR was conducted based on the PRISMA Statement, see File 1 for Checklist [17]. In compliance with the requirements of Brazilian law, we registered our research with the Genetic Heritage Management Council (SisGen, in Portuguese) under number A0AD60B. Our protocol for this review was not previously registered because our research does not analyze directly any health-related outcomes.

### Selection criteria

The following research question guided this review: *Which food plants occurring in the Caatinga biome are strategic for promoting food and nutritional security*?

We selected articles following these eligibility criteria: (i) original articles, published in English, Spanish, or Portuguese, from 2008 to 2020, the year in which we finalized our review; (ii) papers focused on the study of food plants occurring in the Brazilian Caatinga biome.

We set our time frame beginning in 2008 because, in Brazil, the discussion on food biodiversity started to gain visibility from 2009, especially, under the name "*Plantas Alimentícias Não Convencionais*" (UFP, in English). A quick query in *Google Trends* with this term demonstrates the tendency that justifies our clipping. This criterion offered a proxy so that the time frame was not arbitrary.

We also excluded repeated articles and review products.

### Search sources

Between October 2018 and February 2020, we used four databases to perform the search: Web of Science, Medline/PubMed (via the National Library of Medicine), Scopus, and Embrapa Agricultural Research Databases. We used the first three because of their excellent performance in collecting evidence for SR [18]. We added Embrapa's database to gather more Brazilian studies on the topic. Then, we manually checked the reference lists of the articles filtered by the descriptors.

### Search

The research consisted of applying the descriptors in each database. Following the PRISMA guidelines, the search strategy applied to each of the databases is available in the S1 File, attached.

### Study selection

With the assistance of the reference manager *Mendeley*, we organized all records and deleted duplicates. Applying the eligibility criteria previously outlined, one author (MFAM) and one collaborator (LMS) selected the articles individually. Initially, titles and abstracts underwent a first screening, at which point we excluded those that did not meet the selection criteria. In

cases of discrepancies or uncertainties about inclusion, we consulted a second author (MCMJ). Then, we proceeded to a full reading of potentially eligible texts.

## Data extraction

We extracted data from the selected articles into a spreadsheet designed to answer the research question. One author (MFAM) and one collaborator (LMS) were involved in the extraction. We gathered the following information: (i) article data (authors, year of publication, journal); (ii) location of the study and collection of plant material; (iii) objectives; (iv) design; (v) participants (when applicable); (vi) investigated results; (vii) methods; (viii) related results; (ix) quality; and (x) nutritional composition indicators available in the studies. One second author (MCMJ) was responsible to verify the accuracy and scope.

We evaluated the methodological quality of the studies with the support of the following recommendations: Analytical Quality Control (AQC) [19], Strengthening the Reporting of Observational Studies in Epidemiology Statement (STROBE) [20] and the Consolidated Criteria for Reporting Qualitative Research (COREQ) [21].

For the analysis of experimental food studies, we used the AQC, which consists of a checklist of 21 criteria to evaluate reports of chemical analysis. As the identification of plant material is relevant to our analysis and is not in this protocol, we added the item to it. Following the method of Medeiros *et al.* [22] we gave a positive evaluation for this item when the authors identified more than 80% of the taxa at the species level, which the author and her collaborators considered as low risk. Therefore, we analyzed a total of 22 items in the case of experimental studies. In the case of ethnobotanical studies, as there are no consolidated protocols for assessing their overall quality, we chose to adopt and adapt a consolidated protocol, the STROBE, having as reference the objectives of our study. STROBE consists of a checklist of 22 essential items applied to observational epidemiological studies. Again, considering the relevance of the identification of plant material, we added the Medeiros *et al.* reference to the protocol, for a total of 23 items. Finally, we used COREQ to analyze the only qualitative study in our sample. This protocol is intended for the evaluation qualitative research reports that make use of interviews. However, in the absence of a specific instrument for qualitative documentary analysis, we adopted it and evaluated the applicable criteria (18 of 32 items) for the analysis of documents.

After analyzing all the items, the studies received a point for each criterion fulfilled. Based on the grades received, we established three categories for quality assessment: strong—when the study met more than 80% of the criteria; moderate—from 50 to 80%; weak—less than 50%. In cases of studies with mixed methods, we proceeded as follows: we evaluated both phases, with different protocols, and calculated the arithmetic mean. In order to reduce bias in the accumulated evidence, we discarded any study assessed as weak.

## Summary of results

Considering the heterogeneous nature of the included studies, we produced narrative summaries of each of the articles eligible for a full reading.

To survey the plants, we proceeded as follows: initially, we scanned the BFP presented in the studies. We selected plants classified as food in the original studies and described in them at the species level. With a previous list, we checked the scientific nomenclatures using the *Taxonomic Name Resolution Service v 4.0* software. We updated all of them to the accepted nomenclature. We selected plants with occurrence in the Caatinga biome by consulting the *Flora do Brazil 2020* database [23]. We considered the species as native or exotic, taking as reference the "origin" field in *Flora do Brazil*. We considered the occurrence to be in the Caatinga

if in the field "phytogeographic domain" there was a reference to this biome. From the articles, we collected the information to associate with the plants in our final list, such as popular names, edible parts, culinary uses, and nutritional composition indicators, when available.

## Other analyses

For this review, we divided the category of BFP into two: *food plants* and *potentially edible plants*. In the first category, we include those plants reported as food in ethnobotanical or mixed methods studies. The second, on the other hand, includes plants mentioned as edible only in experimental studies, with no mention of their consumption by human groups in the analyzed studies. For our analysis of strategic plants, we considered only the first category, that is, the set of food plants. Of these, we analyzed those that had composition data associated with them.

We emphasize that these two categories of plants may have antinutritional factors and toxic compounds. Our decision to consider only plants with confirmed consumption by human groups was a way of giving an objective reference that indicated a more significant potential for the edibility of the plant.

Using the nutritional indicators provided by the studies—energy (Kcal or KJ) and protein (g, grams)—we analyzed the species that had both higher energy and protein contributions. We obtained this analysis by adding the energy data (converted into Kcal) to the calculation of the energy coming specifically from the protein portion.

The diet of populations experiencing food insecurity in the area of this biome is deficient mainly in protein and also in energy [24]. For this reason, we considered energy and protein together as food markers with the potential to strengthen FNS in the region. We produced a ranking of these plants and analyzed dispersion measures. We highlighted those with values above the average of the set.

## Results

### Study selection

The search in the databases led to the recovery of 318 studies (122 in the Web of Science, 47 in Medline/PubMed, 131 in Scopus and 18 in Embrapa). After excluding 88 duplicates, we considered 230 articles as eligible for the next stage of selection. Based on titles and abstracts, we selected 23 articles for full reading. The articles excluded at this stage were mostly about plants not associated with human consumption, such as plants consumed by animals or with other categories of use, studies on unconventional animals present in human diets, studies on pollination or research on agricultural efficiency of large-scale plantations, such as of soybeans. Of the articles selected for full reading, we excluded eight publications because they did not fit the inclusion criteria. One of the articles, for example, was excluded because it was an analysis of the nutritional composition of a single plant that has no occurrence in the Caatinga biome (*Bombacopsis amazonica* A. Robyns, castanha da chapada). Thus, a total of 15 articles make up this SR. This selection work was carried out by two authors (MFAM and MCMJ) and one collaborator (LMS). Fig 1 shows the study selection process and the related flowchart.

### Study characteristics

Nine of the studies we selected were ethnobotanical, eight of them were observational and cross-sectional [25–32], one was historical [33]. The size of their samples ranged from 15 to 117 people, with an average of 55 participants. The studies were set in communities in the Caatinga area in the Brazilian states of Pernambuco (PE), Paraíba (PB) and Rio Grande do Norte

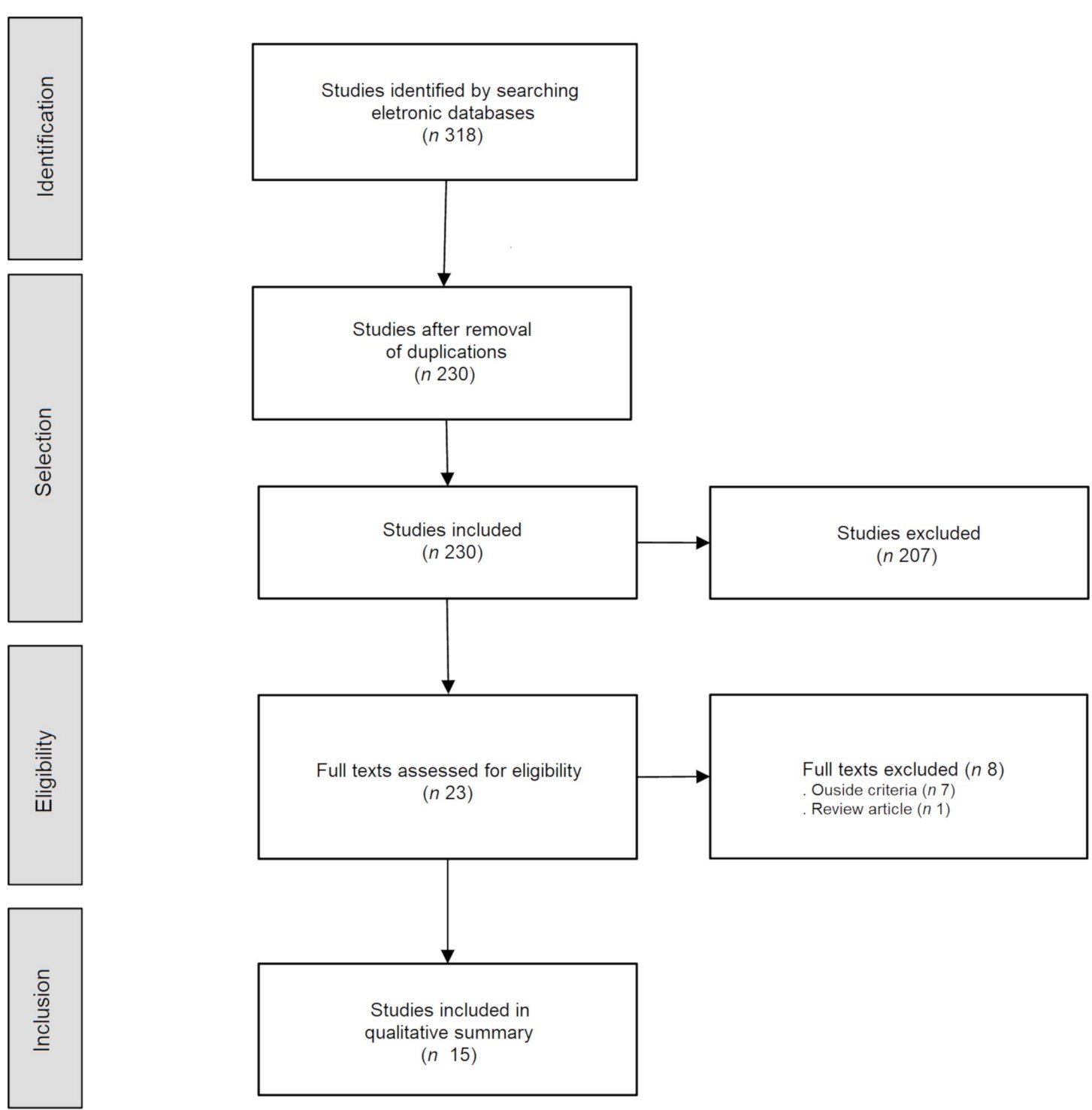

**Fig 1. Flowchart of the study selection process.**

(RN). The historical research was based on the work *Historia Naturalis Braziliae* by Guilherme Piso and Jorge Marcgrave. Considering the scarce and dispersed historical sources of the South American continent, this book is a landmark in scientific studies that aim to make Brazilian flora known.

The randomized experimental studies of analysis of food composition totaled five [24,34–37]. These studies presented indicators of nutritional composition (macronutrients, micronutrients, and bioactive compounds), ranging from 1 to 14 species per study, with an average of approximately five species. All plants analyzed were collected in the field by the authors of the original studies.

One of the studies had a mixed method: an observational phase of ethnobotanical reference (68 participants), and an experimental phase resulting in an analysis of the chemical composition of seven species.

Table 1 provides an overview of the main characteristics of the 15 studies included in this review.

We grouped the results in two parts. The first includes *food plants* and *potentially edible plants*, which consist of the plants mentioned in the studies that met our inclusion criteria. Second, under the title of *strategic food plants*, we present the BFP with a nutritional profile that addresses the main dietary deficiencies in the region.

## Quality analysis

We evaluated the studies of moderate and strong quality. Characteristics of the experimental studies that contributed most to our determination of moderate quality were the omission of the reporting of limits, absence of interlaboratory proficiency tests, and lack of taxonomic identification of flora. Two of the five studies did not report having performed the botanical identification of the analyzed material [34,35]. We did not add nutritional data of these species in our analysis of strategic plants. In ethnobotanical studies, we related moderate quality with omission of study limitations, lack of generalization of the results, and absence of indication of study design. All ethnobotanical studies included botanical identification of species. None of the studies analyzed less than 85% of taxa at the species level. The qualitative study had a strong evaluation. We did not rate any study as weak.

## Food plants and potentially edible plants occurring in the Caatinga

From the studies, we extracted 65 species (Table 2). Some of these plants, occurring in Caatinga, are native to other biomes, such as *Talisia esculenta* (Cambess.) Radlk. (pitomba), *Ilex paraguariensis* L. (erva-mate), *Genipa americana* L. (genipapo), *Inga edulis* Mart. (ingá), and *Piper marginatum* Jacq (capeba). These plants may have been introduced to the region through trade, exchange, or importation and now have their consumption incorporated by local communities. *Dimorphandra gardeneriana* Tul., likewise, although it occurs in the Caatinga, is not native to it. We justify its presence in our data by the fact that its collection happened in the Araripe National Forest, located in a transition zone that presents traces of the Atlantic Forest, Cerrado, and Caatinga.

Native plants corresponded to approximately 89% of the total (*n* 58). The species belong to 22 families, the most frequent being Fabaceae, Euphorbiaceae, Cactaceae, and Arecaceae. The plants most present in the studies were the following: *S. tuberosa* (umbu), *C. jamacaru* (mandacaru), *C. tapia* (trapiá), *H. courbaril* (jatobá), and *S. cearensis* (coco-catolé), being mentioned by seven, six, five, five, and four of the articles, respectively.

## Strategic food plants to promote food and nutrition security in the Caatinga

Of the studies analyzed, six presented data on food composition. In total, they provided analysis of 35 edible items, including varieties, subspecies, and different parts of plants. Only 17 of these items had their consumption also reported in observational ethnobotanical studies.

**Table 1. Characterization of studies regarding biodiverse food plants in Caatinga biome.**

| Study number | Data on publication (authors, year and journal) | Setting | Objective | Design | Participants | Outcomes investigated | Outcomes measurement method | Outcomes | Quality |
|---|---|---|---|---|---|---|---|---|---|
| 1 | Almeida *et al.*, 2016 (34) *Food chemistry* | Mossoró, RN, Brazil | To evaluate the bioactive compounds and the antioxidant potential of the fruit of *Ximenia americana* L. | Randomized experimental study | N/A | Bioactive compounds and antioxidant activity—flavonoids, anthocyanins, carotenoids, vitamin C | Chemical composition analysis of fruits harvested directly from wild plants | Fruits are a potential source of antioxidants, with possible applications in pharmacology, medicine and nutrition | Moderate |
| 2 | Carvalho *et. al.*, 2011 (24) *Journal of Food Composition and Analysis* | Floresta Nacional do Araripe, CE, Brazil | To investigate the food potential of 14 wild legumes from the Caatinga | Randomized experimental study | N/A | Energy, macronutrients, micronutrients and presence of antinutrients—lecithin, trypsin inhibitor, urease–and toxic substances | Analysis of chemical composition of ripe wild seeds collected in dry season | Seeds have nutritional relief equal to or greater than those found in conventional legumes such as beans and soybeans | Strong |
| 3 | Cavalcanti; Bora; Carvajal, 2009 (35) *Cienc. e Tec. de Alimentos* | Santa Luzia, PB, Brazil | To characterize functional properties of the protein isolate of *Cnidoscolus quercifolius* Pohl almonds | Randomized experimental study | N/A | Macronutrients and functional properties (absorption capacity, emulsification and solubility) of the two varieties of the plant | Analysis of chemical composition of ground almonds | High lipid and protein content. Potential for nutritional applications. The thornless variety showed better water and oil absorption capacity | Moderate |
| 4 | Cruz *et al.*, 2014 (25) *Journal of Ethnobiology and Ethnomedicine* | Altinho, PE, Brazil | To analyze participants' perceptions of native edible plants and relate to socioeconomic factors | Ethnobotanical, observational, cross-sectional study | 39 people,> 18 years old, living in one Caatinga rural community | Relationship between the perception of food plants with their use (number of items used) and socioeconomic factors (age, gender, income and occupation) | Semi-structured interviews | Flavor was the positive perception most associated with use; cultural acceptance, negative. Perceptions directly related to age and income | Strong |
| 5 | Cruz; Peroni; Albuquerque, 2013 (26) *Journal of Ethnobiology and Ethnomedicine* | Altinho, PE, Brazil | To relate knowledge, use and management of wild edible plants and socioeconomic factors | Ethnobotanical, observational, cross-sectional study | 39 people,> 18 years old, living in one Caatinga rural community | Relationship between knowledge, use, and management (number of items known, consumed, preparations) with socioeconomic factors (age, gender, income, and occupation) | Semi-structured interviews | Knowledge is related to age, but not to occupation and uses. Association between age and use may indicate abandonment of the resource | Strong |

*(Continued)*

**Table 1.** (Continued)

| Study number | Data on publication (authors, year and journal) | Setting | Objective | Design | Participants | Outcomes investigated | Outcomes measurement method | Outcomes | Quality |
|---|---|---|---|---|---|---|---|---|---|
| 6 | Nascimento *et al.*, 2011 (36) *Food Research International* | Altinho, PE, Brazil | To determine nutritional composition of native Caatinga species | Randomized experimental study | N/A | Energy, macronutrients and bioactive compounds—anthocyanins, flavonoids and carotenoids | Analysis of chemical composition of plants, ripe fruits | Plants with high nutritional potential. The study points out plants of interest for future research on bioactive compounds (e.g., *Sideroxylon obtusifolium* (Roem. & Schult.) T.D. Penn. | Strong |
| 7 | Nascimento *et. al.*, 2012 (38) *Economic Botany* | Altinho, PE, Brazil | Collect ethnobotanical and nutritional data on famine foods | Mixed methods. Phase 1: Ethnobotanical, observational, cross-sectional study. Phase 2: Randomized experimental study | 68 people,> 18 years old, living in two Caatinga rural communities | Phase 1: Relationship between knowledge and socioeconomic factors. Phase 2: Energy, macronutrients and bioactive compounds from the seven main species | Free list and semi-structured interview | There is a difference in knowledge between communities. The data demonstrate the nutritional potential of Caatinga plants. *Mandevilla tenuiflora* (J.C. Mikan) Woodson is indicated for future studies | Strong |
| 8 | Nascimento *et al.*, 2013 (27) *Ecology of Food and Nutrition* | Altinho, PE, Brazil | To compare traditional knowledge regarding food plants in two rural communities in the Caatinga | Ethnobotanical, observational, cross-sectional study | 68 people,> 18 years old, living in two Caatinga rural communities | Relationship between knowledge and use of plants with socioeconomic factors, comparing data from two communities | Free list, semi-structured interview and adapted version of 24h Recall | There is a difference in knowledge between communities. Despite extensive knowledge, native species have low frequency of consumption in communities | Strong |
| 9 | Santos *et al.*, 2009 (28) Economic Botany | Altinho, PE, Brazil | To analyze the contribution of anthropogenic landscapes to providing useful botanical resources | Ethnobotanical, observational, cross-sectional study | 15 people,> 18 years old, living in one Caatinga rural community | Species distribution by categories of use—forage, medicinal, food and timber | Semi-structured interviews and "field herbarium" | The study presents 119 species. Forage was the main category. 10% of the plants have food use, among them *Senegalia bahiensis* (Benth.) Seigler & Ebinger | Strong |

(*Continued*)

**Table 1.** (Continued)

| Study number | Data on publication (authors, year and journal) | Setting | Objective | Design | Participants | Outcomes investigated | Outcomes measurement method | Outcomes | Quality |
|---|---|---|---|---|---|---|---|---|---|
| 10 | Santos et al., 2014 (29) Economic Botany | Crato, CE, Brazil; Caruaru, PE, Brazil | To investigate the usefulness of invasive native and exotic plants for residents of two different communities | Ethnobotanical, observational, cross-sectional study | 106 people,> 18 years old, living in two Caatinga rural communities | Relate species considered invasive (native and exotic) with their local perception of usefulness | Semi-structured interviews and plot method for vegetation sampling | 55 of the 56 local species considered invasive are considered useful. Participants mentioned 12% of plants as food, among them Passiflora cincinnata Mast | Strong |
| 11 | Ferraz et al., 2012 (30) Bosque | Floresta, PE, Brazil | To know the types of use of woody vegetation made by indigenous family farmers | Ethnobotanical, observational, cross-sectional study | 30 people,> 18 years old, living in one Caatinga rural community | Categories of use of woody species —food, fodder, fuel, construction | Participant observation and semi-structured interviews | 27 species identified. Forage was the main use category. 11% of the plants are mentioned as food, among them Croton blanchetianus Baill | Moderate |
| 12 | Juvik et al., 2017 (37) Molecules | Petrolina, PE, Brazil | To identify non-polar constituents of Bromelia laciniosa Mart. ex Schult. & Schult. f., Neoglaziovia variegata (Arruda) Mez and Encholirium spectabile Mart. ex Schult. & Schult.f. | Randomized experimental study | N/A | Fatty acids and their derivatives, very long chain alkanes, vitamins ($\alpha$ and $\beta$-tocopherol), triterpenoids and derivatives | Analysis of chemical composition of plants | Plants with high nutritional potential. Highlight for the presence of vitamin E and phytosterols with potential beneficial health effects | Strong |
| 13 | Medeiros; Albuquerque, 2014 (33) Journal of Ethnobiology and Ethnomedicine | N/A | To list the food plants described in História Naturalis Braziliae (Piso and Marcgrave, 17th century) with a focus on the Caatinga | Ethnobotanical, historical, descriptive study | N/A | Taxonomic classification, identification of plant parts, forms of consumption and verification of use over time | Historical document analysis and databases search | The use of 80 food species is recommended, such as Spondias tuberosa Arruda and Cereus jamacaru DC. Some lack nutritional studies | Strong |
| 14 | Nunes et al., 2018 (31) Journal of Ethnobiology and Ethnomedicine | São Mamede, PB, Brazil; Lagoa, PB, Brazil; Itaporanga, PB, Brazil | To investigate the knowledge of food plants in three communities, comparing communities and gender | Ethnobotanical, observational, cross-sectional study | 117 indigenous farmers,> 18 years old, living in three Caatinga rural communities | Comparison of knowledge of native plants in the three communities and their relationship with socioeconomic factors | Semi-structured interviews | 26 food species are mentioned, especially Spondias tuberosa Arruda. Knowledge of residents of the three communities is low | Strong |

(Continued)

**Table 1.** (*Continued*)

| Study number | Data on publication (authors, year and journal) | Setting | Objective | Design | Participants | Outcomes investigated | Outcomes measurement method | Outcomes | Quality |
|---|---|---|---|---|---|---|---|---|---|
| 15 | Roque; Loiola, 2013 (32) *Revista Caatinga* | Caicó, RN, Brazil | To identify the main categories of use of native plants in a rural community in the Caatinga | Ethnobotanical, observational, cross-sectional study | 23 local experts, > 35 years, living in one Caatinga rural community | Categories of use of native species—medicinal, food, timber, mystical, fuel, forage, domestic use | Semi-structured and structured interviews | The use of 69 species has been described. Medicinal potential related to almost 90% of the plants.11% were food, with emphasis on *Ziziphus joazeiro* Mart. and *S. obtusifolium* | Moderate |

These data correspond to 15 species of food plants since *P. gounellei* (xique-xique) and *P. pachycladus* sub. Pernambucoensis (facheiro) had both cladodes and fruits analyzed and consumed.

The indicators consisted mostly of energy data, macronutrients (protein, fat, carbohydrate), dietary fiber, micronutrients (vitamin C, vitamin E, potassium, sodium, calcium, magnesium, iron, zinc, manganese, copper, chromium, molybdenum), and bioactive compounds (carotenoids, flavonoids, anthocyanins). We compiled data for the items that had energy and protein indicators available in the studies we reviewed. The complete list of these plants is available in Table 3.

All species are native. Of this group of plants, *D. grandiflora*, *H. courbaril*, *S. cearensis*, *L. ferrea*, and *S. obtusifolium* have higher energy and protein values than the group average (Fig 2).

Three of these five species that lead the ranking are Fabaceae. Based on their nutritional content, we highlight the value of legumes, within the set of strategic plants, as species that can significantly contribute to improving the pattern of diets in the region. The plants that lead the ranking are in Fig 3.

Legend reads clockwise from upper left: Flower, leaves, and seeds of *D. grandiflora*, by Michelle Jacob. Pods of *L. ferrea* by Natalia Araújo. Fruit of *S. obtusifolium* by Gildásio Oliveira. Nut of *S. cearensis* by Michelle Jacob. *H. courbaril* by Neide Rigo.

In energy terms, the most significant contribution is from *H. courbaril* with 431 Kcal for every 100 g of seeds. In protein, *L. ferrea* is ranked first, with approximately 43% protein in its seeds. In total energy and protein, top ranked is *D. grandiflora*.

*D. grandiflora* is cited by Teixeira *et al.* [38] in her research on famine foods, that is, plants used as food in times of scarcity. In her study, six people mentioned the use of the seeds of this species in periods of extreme drought to produce flour, prepared as couscous. The consumption of 100 g of *mucunã* seeds provides approximately 62% and 18% of the daily protein and energy requirements, respectively [24].

Teixeira *et al.* [27], Cruz *et al.* [25,26] and Nunes *et al.* [31] reported the consumption of *H. courbaril*, in rural communities in the semiarid regions of Pernambuco and Paraíba. The authors mentioned the use of fresh fruit, especially in the form of flour. The intake of 100 g of this fruit contributes to about 22% of the daily requirements for both protein and energy [24].

*L. ferrea* is also reported by Nunes *et al.* [31], who described the use of seeds in the form of flour in rural communities in the semiarid region of Paraíba. For every 100 g of seeds, the protein supply is 42.7 g, which corresponds to more than 85% of the daily recommendation. In

**Table 2. Synthesis of food plants and potentially edible plants occurring in the Caatinga.**

| Number | Scientific name | Popular name | Reporting studies | Origin | Edible part | Culinary uses |
|---|---|---|---|---|---|---|
| ANACARDIACEAE | | | | | | |
| 1 | *Commiphora leptophloeos* (Mart.) J.B. Gillett | umburana | et (#14) | native | fruit | raw (spice) |
| 2 | *Spondias tuberosa* Arruda | umbu; umbuzeiro; imbu | et (#4 #5 #8 #9 #11 #13 #14) | native | fruit; tuber; leaf | raw (juice); cooked (*umbuzada*[i]); preserve (jam) |
| APOCYNACEAE | | | | | | |
| 3 | *Mandevilla tenuifolia* (J.C. Mikan) Woodson | manofê | et (#4 #5 #8); mx (#7) | native | tuber | raw (salad; juice); preserve (pickles) |
| AQUIFOLIACEAE | | | | | | |
| 4 | *Ilex paraguariensis* L. | erva-mate | et (#8) | native | leaf | na |
| ARECACEAE | | | | | | |
| 5 | *Copernicia prunifera* (Mill.) H.E. Moore | carnaúba | et (#14 #15) | native | fruit | raw |
| 6 | *Syagrus cearensis* Noblick | coco-catolé; catolé; coco-babão | et (#4 #5 #8); fc (#6) | native | fruit | na |
| 7 | *Syagrus coronata* (Mart.) Becc. | licuri; licurizeiro | et (#13) | native | seed | na |
| 8 | *Syagrus oleracea* (Mart.) Becc. | coco-catolé | et (#14) | native | fruit | raw |
| BORAGINACEAE | | | | | | |
| 9 | *Varronia globosa* (Jacq.) Kunth | moleque-duro | et (#8) | native | fruit | na |
| BROMELIACEAE | | | | | | |
| 10 | *Bromelia laciniosa* Mart. ex Schult. & Schult.f. | macambira; macambira-roxa; macambira-de-porco | et (#8); fc (#12) | native | leaf | cooked (flour/bread) |
| 11 | *Encholirium spectabile* Mart. ex Schult. & Schult.f. | macambira-de-flexa; macambira-de-pedra | mx (#7); et (#8); fc (#12) | native | leaf | cooked (flour/couscous[ii]) |
| 12 | *Neoglaziovia variegata* (Arruda) Mez | caroá | fc (#12) | native | leaf; fruit | leaf: cooked (flour/couscous[ii]); fruit: cru |
| CACTACEAE | | | | | | |
| 13 | *Cereus jamacaru* DC. | mandacaru; cardeiro; babão | et (#4 #5 #8 #13 #15); fc (#6) | native | cladode; fruit | cladode: cooked; fruit: raw; cooked; preserve |
| 14 | *Melocactus zehntneri* (Britton & Rose) Luetzelb. | coroa-de-frade | et (#8) | native | fruit | na |
| 15 | *Pilosocereus gounellei* (F.A.C.Weber) Byles & Rowley | xique-xique | fc (#6); mx (#7); et (#8 #15) | native | cladode; fruit | cooked (flour/couscous[ii]); baked |
| 16 | *Pilosocereus pachycladus* subsp. pernambucoensis (Ritter) Zappi | facheiro | et (#4 #5 #8); fc (#6) | native | cladode; fruit | raw; preserve (candy) |
| 17 | *Tacinga inamoena* (K.Schum.) N.P. Taylor & Stuppy | cumbeba | fc (#6); et (#8) | native | cladode; fruit | raw; preserve (jam) |
| CAESALPINIACEAE | | | | | | |
| 18 | *Bauhinia cheilantha* (Bong.) Steud. | mororó | et (#8) | native | leaf; seed | na |
| CAPPARACEAE | | | | | | |
| 19 | *Capparis flexuosa* (L.) L. | feijão-de-boi | et (#8) | native | seed | na |
| 20 | *Crataeva tapia* L. | trapiá | et (#4 #5 #8 #13 #14) | native | fruit | raw |
| 21 | *Neocalyptrocalyx longifolium* (Mart.) Cornejo & Iltis | incó | et (#4 #5 #8) | native | fruit | na |
| CELASTRACEAE | | | | | | |
| 22 | *Monteverdia rigida* (Mart.) Biral | bom-nome | et (#8) | native | fruit | na |
| EUPHORBIACEAE | | | | | | |
| 23 | *Cnidoscolus quercifolius* Pohl | favela-branca; faveleira | fc (#3); et (#11 #14) | native | seed | cooked (flour) |
| 24 | *Cnidoscolus urens* (L.) Arthur | urtiga; cansanção | et (#8) | native | seed | na |

(*Continued*)

**Table 2.** (Continued)

| Number | Scientific name | Popular name | Reporting studies | Origin | Edible part | Culinary uses |
|---|---|---|---|---|---|---|
| 25 | *Croton blanchetianus* Baill | marmeleiro | et (#11) | native | na | na |
| 26 | *Manihot dichotoma* Ule. | maniçoba | mx (#7); et (#8) | native | raw | cooked (flour/beiju[iii]) |
| 27 | *Manihot glaziovii* Müll.Arg. | purnunça; maniçoba | mx (#7); et (#8 #13) | native | raw | cooked (flour/beiju[iii]) |
| 28 | *Ricinus communis* L. | mamona; azeite | et (#10) | exotic | leaf; flower; fruit; seed | na |
| FABACEAE | | | | | | |
| raw | *Amburana cearensis* (Allemão) A.C. Sm. | cumaru | et (#14) | native | fruit | raw |
| 30 | *Cajanus cajan* (L.) Huth. | feijão-guandu; feijão-andu | et (#13) | exotic | seed | cooked |
| 31 | *Dimorphandra gardneriana* Tul. | fava-d'anta | fc (#2) | native | seed | na |
| 32 | *Dioclea grandiflora* Mart. ex Benth | mucunã | mx (#7) | native | seed | cooked (flour/couscous[ii]) |
| 33 | *Dioclea megacarpa* Rolfe | mucunã; olho-de-boi | fc (#2) | native | seed | na |
| 34 | *Enterolobium contortisiliquum* (Vell.) Morong | orelha-de-macaco; orelha-de-negro | fc (#2) | native | seed | na |
| 35 | *Erythrina velutina* Willd. | mulungu | fc (#2) | native | seed | cooked |
| 36 | *Hymenaea courbaril* L. | jatobá | #2 #4 #5 #8 #14 | native | fruit | raw (flour) |
| 37 | *Inga edulis* Mart. | ingá | et (#8) | native | fruit | na |
| 38 | *Lablab purpureus* (L.) Sweet | feijão-cabricuço; mandatia | #13 | exotic | fruit; flower | cooked; raw |
| 39 | *Libidibia ferrea* (Mart. ex Tul.) L.P. Queiroz | jucá; pau-ferro | #14 #2 | native | seed | cooked (flour) |
| 40 | *Lonchocarpus sericeus* (Poir.) Kunth ex DC. | ingá | fc (#2) | native | seed | na |
| 41 | *Parkia platycephala* Benth. | visgueiro | fc (#2) | native | seed | na |
| 42 | *Phaseolus lunatus* L. | fava | et (#8) | exotic | seed | na |
| 43 | *Piptadenia moniliformis* Benth. | catanduva | fc (#2) | native | seed | na |
| 44 | *Pterogyne nitens* Tul. | madeira-nova | fc (#2) | native | seed | na |
| 45 | *Senegalia bahiensis* (Benth.) Seigler & Ebinger | espinheiro | et (#8) | native | fruit | na |
| 46 | *Senna obtusifolia* (L.) H.S.Irwin & Barneby | mata-pasto | fc (#2) | native | seed | na |
| 47 | *Senna occidentalis* (L.) Link | manjiroba | #9 | native | na | na |
| 48 | *Senna rugosa* (G.Don) H.S.Irwin & Barneby | lagarteiro | fc (#2) | native | seed | na |
| 49 | *Caesalpinia bracteosa* Tul. | catingueira; catinga-de-porco | fc (#2) | native | seed | na |
| MYRTACEAE | | | | | | |
| 50 | *Myrciaria cauliflora* (C. Martius) O. Berg | jabuticaba | et (#8) | native | fruit | na |
| 51 | *Psidium schenckianum* Kiaersk. | pirim; araçá-do-cerrado | et (#4 #5 #8); fc (#6) | native | fruit | na |
| OLACACEAE | | | | | | |
| 52 | *Ximenia americana* L. | ameixa-do-mato; ameixa-silvestre | fc (#1); et (#14) | native | fruit | raw (juice) |
| PASSIFLORACEAE | | | | | | |
| 53 | *Passiflora cincinnata* Mast. | maracujá-do-mato; maracujá-brabo; maracujá-de-boi; murucujá | et (#8 #10 #13) | native | fruit; flower; leaf; seed | na |
| 54 | *Passiflora foetida* L. | maracujá-de-estralo; canapú; maracujá; maracujá-do-mato | et (#8 #9 #14 #15) | native | fruit | raw |
| PIPERACEAE | | | | | | |
| 55 | *Piper marginatum* Jacq. | capeba | et (#13) | native | fruit | raw (spice) |
| PORTULACACEAE | | | | | | |

*(Continued)*

**Table 2.** (Continued)

| Number | Scientific name | Popular name | Reporting studies | Origin | Edible part | Culinary uses |
|--------|-----------------|--------------|-------------------|--------|-------------|---------------|
| 56 | *Portulaca oleracea* L. | beldroega; bredoégua; caaponga | et (#10 #13) | exotic | leaf; stalk; flower | cooked |
| RHAMNACEAE | | | | | | |
| 57 | *Ziziphus joazeiro* Mart. | juá; juazeiro | fc (#6); et (#8 #14 #15) | native | fruit | raw |
| RUBIACEAE | | | | | | |
| 58 | *Genipa americana* L. | genipapo; ianupaba; ienipapo | et (#13) | native | fruit | raw; preserve (liquor) |
| SAPINDACEAE | | | | | | |
| 59 | *Talisia esculenta* (Cambess.) Radlk. | pitomba; nhua | et (#8 #13) | native | fruit | na |
| SAPOTACEAE | | | | | | |
| 60 | *Sideroxylon obtusifolium* (Roem. & Schult.) T.D.Penn. | quixabeira; quixaba | fc (#6); et (#8 #14 #15) | native | fruit | raw |
| SOLANACEAE | | | | | | |
| 61 | *Physalis angulata* L. | canapú | et (#15) | exotic | na | na |
| 62 | *Solanum agrarium* Sendtn. | gogóia; melancia-da-praia | et (#9 #15) | native | na | na |
| 63 | *Solanum americanum* Mill. | erva-moura; maria-pretinha | et (#10) | native | leaf; fruit | na |
| 64 | *Solanum rytidoandrum* Sendtn | jurubeba; jurubeba-de-espinho; espinho | et (#10) | exotic | leaf | na |
| VERBENACEAE | | | | | | |
| 65 | *Lantana camara* L. | chumbinho | et (#9 #10) | native | stalk; flower; leaf; fruit | na |

#: revised study number (see Table 1)

na: not available

et: ethnobotanical study

fc: food composition study

mx: mixed method study.

[i] Typical drink from the Northeast region, prepared with the fruit of the cooked *umbu*, mixed with milk, and sugar.

[ii] Sweet or salty dish prepared with steamed vegetable flour.

[iii] Cooked pasta dish prepared from vegetable flour of the genus Manihot.

energy, the contribution is around 12% [24]. These data highlight the potential nutritional value of plants in this biome.

The other two species in our ranking are *S. cearensis* and *S. obtusifolium*. Cruz *et al*. [25,26] and Nascimento *et al*. [27,36] reported *S. cearensis* consumption in rural communities of Caatinga in the state of Pernambuco. The edible part is the fruit, without specification of culinary use. Analysis of its endosperm reveals a contribution of 394 kcal for every 100 g of material analyzed, which corresponds to almost 20% of the daily energy recommendation.

Several ethnobotanical studies in the semiarid region of Pernambuco, Paraíba, and Rio Grande do Norte [27,31,32,36] report the consumption of fresh fruit of *S. obtusifolium*. Its energy, 212 kcal per 100 g, or approximately 11% of the daily recommendation, positions it as a potential species to integrate into FNS programs in the region [36].

Although our analysis focuses on energy and protein, several plants on our list are significant sources of antioxidants, such as *S. obtusifolium* analyzed by Teixeira *et al*. [39]. The same author and her colleagues in a later study [38] analyzed the flavonoid content of *D. grandilflora* and found significant quercetin values. These studies also highlight the content of bioactive compounds from fruits of Cactacea species, such as *P. pachycladus* subsp. pernambucoensis (facheiro), *T. inamoena* (cumbeba), and *P. gounellei* (xique-xique).

**Table 3. Nutritional data of strategic food plants to promote food and nutrition security in the Caatinga.**

| Food plant | Part analyzed | Energy (Kcal)[i] | Protein (g)[i] | Protein energy (Kcal) | Protein energy + energia (Kcal) | Data source |
|---|---|---|---|---|---|---|
| *Dioclea grandiflora* Mart. ex Benth | seed | 367 | 30,90 | 124 | 491 | #7 |
| *Hymenaea courbaril* L. | fruit | 431 | 10,9 | 44 | 475 | #2 |
| *Syagrus cearensis* Noblick | endosperm | 394 | 8,95 | 36 | 430 | #6 |
| *Libidibia ferrea* (Mart. ex Tul.) L.P.Queiroz | seed | 239 | 42,7 | 171 | 410 | #2 |
| *Sideroxylon obtusifolium* (Roem. & Schult.) T.D.Penn. | fruit | 212 | 2,86 | 11 | 223 | #6 |
| *Psidium schenckianum* Kiaersk. | fruit | 125 | 1,64 | 7 | 132 | #6 |
| *Encholirium spectabile* Mart. ex Schult. & Schult.f. | leaf | 125 | 0,70 | 3 | 127 | #7 |
| *Pilosocereus gounellei* (F.A.C.Weber) Byles & Rowley | fruit | 102 | 2,65 | 11 | 113 | #6 |
| *Ziziphus joazeiro* Mart. | fruit | 96 | 2,19 | 9 | 105 | #6 |
| *Manihot dichotoma* Ule. | root | 104 | 0,10 | 0 | 104 | #7 |
| *Manihot glaziovii* Müll.Arg. | root | 80 | 1,01 | 4 | 84 | #7 |
| *Tacinga inamoena* (K.Schum.) N.P.Taylor & Stuppy | fruit | 72 | 0,97 | 4 | 76 | #6 |
| *Pilosocereus pachycladus* (Ritter) Zappi subsp. pernambucoensis | fruit | 67 | 2,10 | 8 | 76 | #6 |
| *Cereus jamacaru* DC. | fruit | 64 | 1,80 | 7 | 71 | #6 |
| *Mandevilla tenuifolia* (J.C. Mikan) Woodson | root | 63 | 0,70 | 3 | 66 | #7 |
| *Pilosocereus gounellei* (F.A.C.Weber) Byles & Rowley | cladode | 28 | 0,40 | 2 | 30 | #6 |
| *Pilosocereus pachycladus* (Ritter) Zappi subsp. pernambucoensis | cladode | 25 | 0,25 | 1 | 26 | #6 |

#: revised study number (see Table 1).

[i] Reference Daily Intake– 2000 kcal and 50g of protein (39)

## Discussion

The main objective of this SR was to identify plants occurring in the Caatinga that could be strategic in the promotion of FNS. For this, we listed and characterized the species occurring in the biome and produced a list of strategic plants with nutritional data. Based on our analysis, we highlighted the energy and protein potential of native legumes.

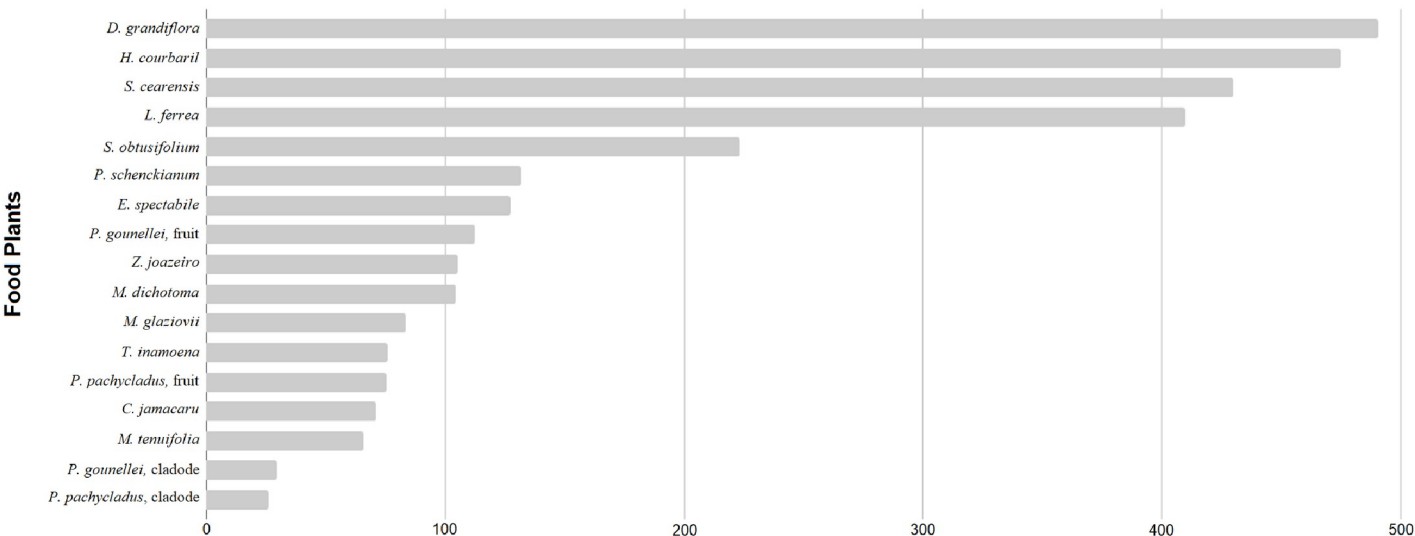

**Fig 2. Strategic biodiverse food plants to promote food and nutrition security.**

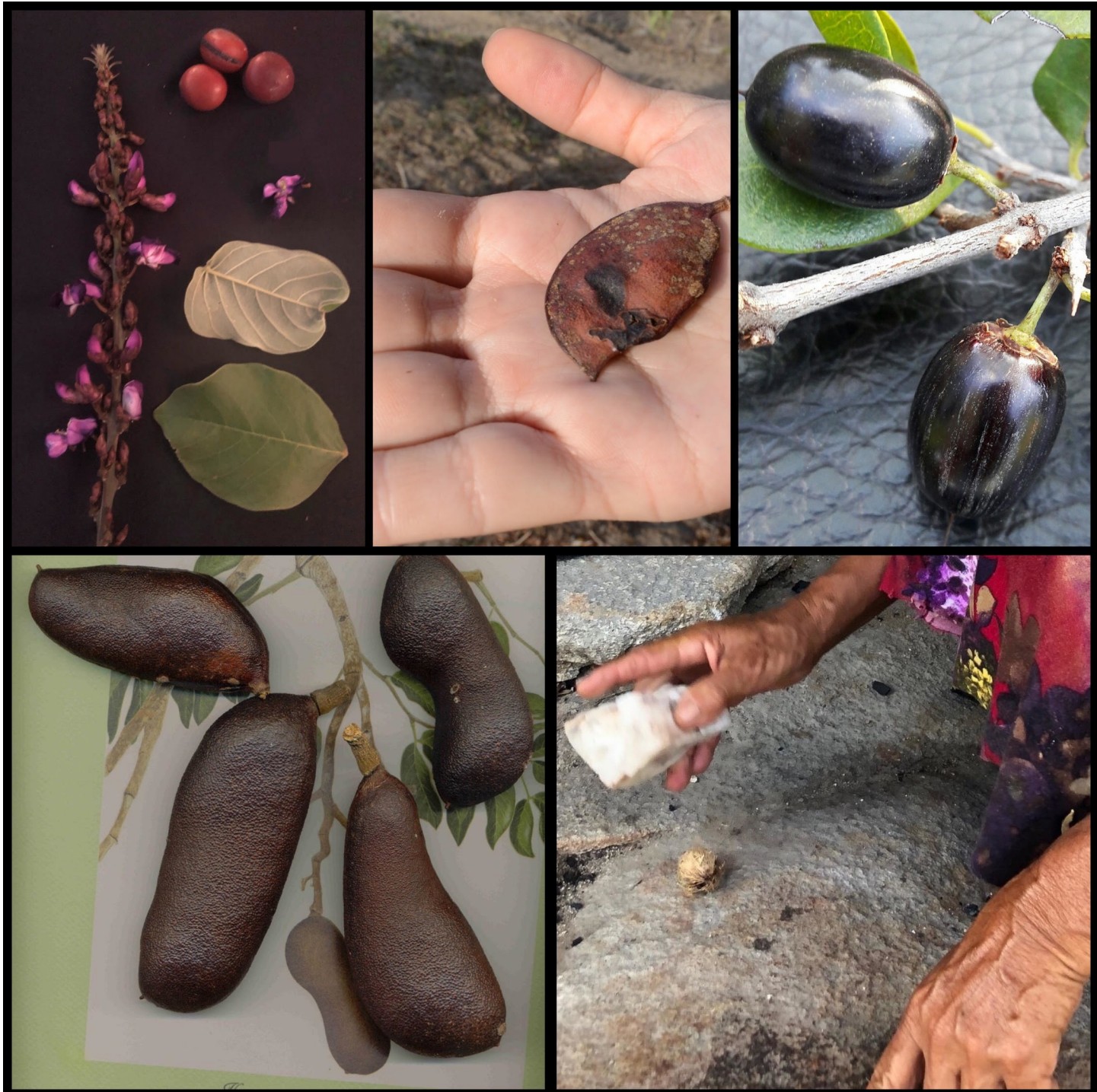

**Fig 3. Top five food plants of strategic species ranking.**

We believe that the species richness surveyed in this review (*n* 65) underestimates the potential of BFP in the Caatinga. A search of the *Flora do Brazil* database in February 2020, for a listing of Angiosperms occurring in this environment, returns 4,890 species. Consider the fact, for example, that of the eight ethnobotanical studies that we analyzed, five of them were

performed in the same community (Carão, Altinho, PE). Our data possibly underestimate the state of food biodiversity in the biome, because, in addition to having the same community as a source of information, they do not include information on BFP from the provinces of Maranhão, Piauí, Sergipe, and northern Minas Gerais, which are also areas of Caatinga. These geographic gaps strongly suggest the need for more ethnobotanical studies in the region.

The fact that most species are native indicates they could be positively related to sustainable diets. We have at least five arguments to sustain this thesis. First, from the environmental point of view, the native species listed as strategic are recognized for their ability to cope with drought, requiring few water resources. The consumption of Euphorbiaceae roots, for example, is typical in scenarios of water scarcity, since these plants can remain intact in the soil for a long time, even in periods of drought [40]. Ecophysiology studies done with palm trees (Arecaceae) also describe species tolerant to water stress [41,42]. Similarly, Fabaceae have several strategies for adapting to drought, including shortening the growth period, maintaining high tissue water potential, reducing water loss, and improving water uptake [43]. About Fabaceae, we realized that the three legumes leading our ranking are arboreal. Dubeux Júnior *et al.* [44] state that in the current climate change condition, tree legumes are an essential component for strengthening FNS. One of the authors' arguments is the resilience of these species, which tend to be more perennial than most herbaceous legumes. This characteristic is relevant in areas of the Caatinga, recognized by their long periods of rain scarcity.

Consequently, our second argument is that positive economic effects, such as saving water inputs, tend to enhance the local supply of food to local communities, to facilitate the opening of local-based markets, and to increase economic resilience in family farmers [45].

Third, nutritionally, low-diversity diets are a challenge to public health at a global level. As a matter of availability, native plants have the potential to increase the diversity of food in local communities [10]. Besides, encouraging conscious diets including native plants also is supported by an environmental argument: It is a path to make BFP known and, thus, increase possibilities for their conservation [46].

Fourth, there is a cultural reason. Native plants are part of the cultural heritage of local populations. Preserving them means safeguarding the traditional knowledge associated with these plants and, consequently, cultural diversity [47].

Finally, we add the fifth argument, which is political: food sovereignty. To food sovereignty, native plants have both cultural and genetic heritage roles. Safeguarding native species, knowledge associated with them, and biological property is part of the process of people taking control of their food heritage [48].

We do not disregard, however, that the introduction of exotic species may have a rational basis, as proposed by Albuquerque *et al.* [49] with the diversification hypothesis. This hypothesis posits that local systems can introduce exotic plants to expand the repertoire of communities. In the case of food, for example, naturalized exotic species, in the absence of native species, can be rationally included to expand the diversity of diets and, consequently, their quality.

The Fabaceae, among the plants we analyzed, are especially good for their nutritional quality. We add two points to the discussion on food legumes and nutrition: antinutritional factors and protein quality.

First, presence of antinutritional factors (such as glucosinolates, trypsin inhibitors, hemagglutinins, tannins, phytates, and gossypol) is one of the biggest limitations on the use of legumes by humans [50,51]. In the analysis by Carvalho *et al.* [24] antinutritional and toxic factors detected in legumes are not a problem for humans if the seeds are correctly processed. They also argue that similar factors are present in popular legumes (e.g., beans and soybeans) before the application of heat treatment. However, we argue that there are other

phytochemicals with high toxicity not tested by the authors (e.g., alkaloids, cyanogenic glycosides) that should be considered. For example, *D. grandiflora* is one of the plants that appear in Carvalho's study as potentially safe after processing. However, local communities [49] claim that the consumption of this species can cause intestinal problems, or even death, due to its toxicity. According to residents, in addition to heat treatment, one of the ways to mitigate, and even eliminate these effects, is washing the flour several times before using it in food processing [38]. Grant *et al*. [52] affirm that the exhaustive dialysis procedure of *mucunã*'s seed flour helps to eliminate soluble components of small molecular weight potentially related to its toxicity. These data show the relevance of new studies to list compounds related to toxicity, as well as studies to gather processing techniques used in local food systems (e.g., bleaching, cooking, washing, fermentation, and dehydration, among others) to inactivate or reduce species' toxicity [53–55]. We do not recommend the consumption of *D. grandiflora* and *L. ferrea* until new research provides additional evidence.

A second point, which concerns protein quality: plant-based proteins have a lower anabolic potential than those animal-based [56]. Two strategies can be useful to ensure the intake of essential amino acids in plant-based diets: increase the intake of proteins and improve the quality of those present in the diet [57]. The Acceptable Macronutrient Distribution Range (AMDR) suggests that protein intake should provide between 10% and 35% of the daily dietary calorie recommendation. Thus, a plant-based diet should be more aligned with the upper limit of this recommendation, that is, 35%. To improve the quality of the ingested proteins, one of the possibilities is to expand the diversity of plant proteins, blending species with different limiting amino acids [57]. Diets that include a variety of vegetable protein sources consistently demonstrate nutritional adequacy when it comes to providing sufficient amounts of essential amino acids [58, 59]. Because of these characteristics, the Food and Agriculture Organization recommends that legumes should be consumed daily as part of a healthy diet, which simultaneously prevents undernutrition, obesity, and non-communicable diseases [60].

In addition to nutritional quality and the ability to adapt to water scarcity, we added three other advantages that serve to consolidate the potential of legumes in the Caatinga region.

First, legumes' potential to fix nitrogen in the soil enriches it without the need for commercial chemical fertilizers and, consequently, offers economic and environmental advantages for sustainable agriculture [61]. Second, legumes are related to smaller land footprints when compared to vegetable proteins and, besides, they do not reduce their nutritional potential when stored for long periods. Thus, they can simultaneously reduce indicators of food loss and food waste [62]. Finally, the third reason is that legumes allow for various culinary applications, ranging from stews and flours to dumplings, as is the case of *acarajé* (fried dumpling made with beans, *Vigna unguiculata* (L. Walp., common in Bahia, Brazil), and desserts, like *paçoca* (Brazilian candy made with peanuts, *Arachis hypogaea* L., common in the Southeast region).

The other two species in our ranking are *S. cearensis* and *S. obtusifolium*, which in addition to their energy content, contribute with other nutrients. *S. cearensis* shows, for example, its fat profile of 69.33 g or approximately 107% of the daily intake recommendation [63, 39]. Also, the species, being a typical palm that grows in semiarid regions, is a strategic source of provitamin A for rural communities in the Caatinga [64]. Each 100 g of the endosperm contains 456 mcg of REA, which corresponds to approximately 91% of the daily needs established for women and 73% for men [65]. Besides, the content of bioactive compounds of *S. obtusifolium* corresponds to almost 12 times the amount of beta-carotene in *acerola* (*Malpighia glabra* L.), 83% of quercetin present in the same portion of red onions (*Allium cepa* L.), and ten times the anthocyanins content found in *jabuticaba* (*Myrciaria cauliflora* (Mart.) O. Ber) [66–68]. These data reinforce the potential of native Brazilian flora as a source of nutrients and bioactive compounds.

However, studies report a decrease in consumption and knowledge associated with these plants [25–27,38]. We list here some reasons to explain this phenomenon. First, an increase in temperature and a decrease in precipitation in the Caatinga is associated with the rise of income transfer by government programs that, in turn, boosts the popularity of acquiring processed food in supermarkets, leading to a decrease in the availability of food produced from local plants [27, 69]. Second, the dynamics of globalized agri-food systems tend to uniformity: monocultures, concentration of supply and distribution centers, and monotonous dietary patterns [70]. Thus, the closer these communities are to urban centers, the higher their permeability to this process of standardization [71]. Third, there is a stigma related to these plants as "poor people's food" [25,38]. Cruz *et al.* [25], in her study of the perception about native plants in Pernambuco, associated the consumption of these species with low social status. Thus, there is a stigma related to their use. Other studies report the same stigma [38,72,73]. Together, these factors collaborate to increase the presence of processed and ultra-processed food products in people's diets, with negative impacts on their nutritional status [1,74]. These arguments indicate the role of intersectoral FNS policies, which involve not only income transfer, but access to food and nutrition education programs and policies to promote family farming and local markets. The FNS implied by resilience and food sovereignty of local food systems is a matter of intersectoral policies.

We started the paper, considering, above all, nutritional aspects of the plants listed in the studies we analyzed. Thus, our ranking was created based on biological criteria. Human diets, however, are complex and also involve cultural factors. Diets are located between nature and culture, as the anthropologist Claude Lévi-Strauss asserted [75]. Considering this fact, we suggest that campaigns to promote the use of BFP cannot be based only on the plant's nutrient profiles. They should also consider if the plants are recognized and appreciated by local culture. In this sense, we highlight *H. courbaril* and *S. cearensis* as crucial plants, as they are simultaneously in the ranking of strategic plants and among the most cited in studies.

Finally, we add that the promotion of these plants, in the context of food systems, also depends on broader actions. Some of them are to integrate food biodiversity in government policies and programs, to provide agricultural incentives to family farmers, to register traditional knowledge, to promote sustainable use of species with consumers, and to foster multidisciplinary research [76].

## Limitations

First, the number of plants we examined was reduced by including only species with composition data available in the studies we reviewed. To analyze the others, we could have performed Food Matching, a strategy to match composition data available in tables and other databases. However, we made the decision not to perform it and focus our analysis on plants with data available in the review, considering that the nutritional composition varies depending on environmental and cultural factors (*terroir*, climate, soil) [77]. The authors who analyzed the plants in the original studies collected them in the Caatinga, which allows us to recognize their real contribution to local communities in nutritional terms. Another limitation was the lack of specific protocols for assessing the overall quality of observational ethnobotanical studies and qualitative studies focusing on documentary analysis. To address this limitation, we used consolidated protocols, and adapted them. Finally, the third limitation is the fact that we set cutoff points in the quality assessment, due to the lack of consensus in the literature on the issue. In order to minimize biases, we analyzed review studies in relevant databases to adopt approximate assessment categories used in other studies.

## Conclusion

Based on this review, the food resources available in the Caatinga offer diversity and quality to address the challenges posed in the characteristics of the region and by current food systems. We suggest that scientific researchers focus their efforts on Fabaceae, especially tree legumes, which, due to their physiological, nutritional, and culinary qualities, simultaneously articulate human and environmental health, economic resilience, and sustainable agriculture. We advocate the recognition of these plants as strategic in building a research agenda on food biodiversity.

We highlight the need for researchers to collect information on culinary uses of species in ethnobotanical studies on food plants. In our analysis, half of the studies did not present this data. This information will make it possible for us to advance collectively in the discussion about antinutritional factors and toxicity associated with these plants. In this sense, we also emphasize the need for ethnoculinary studies with a focus on legumes.

The consumption of BFP is one of the pillars of sustainable diets. We hope that the data presented in this review can encourage the study of these plants. Thus, provided with evidence about their potential and safety, we will be able to support the formulation of food and agriculture policies, as well as sustainable diet guidelines based on local plants.

## Supporting information

**S1 Checklist. PRISMA 2009 checklist.**
(DOC)

**S1 File. Research strategy for systematic review.**
(DOCX)

## Acknowledgments

To Luciana Medeiros Souto (LMS) for her collaboration in data collection. To Ivanilda Soares Feitosa for the support in checking occurrence data. To Samile Laura Dias Barros, Alice Medeiros Souza, Giovanna Guadalupe Cordeiro de Oliveira, André Luiz Dadona Benedito, Fillipe Oliveira Pereira, and Elias Jacob de Menezes Neto for having collaborated with the construction of the methodology. We also thank Thiago Perez Jorge, Adriana Monteiro de Almeida, Célia Marcia Medeiros Morais, Clélia de Oliveira Lyra, Severina Carla Vieira Cunha Lima, and Fernanda Antunes Carvalho for exchanging ideas in the construction of the umbrella project. To teacher Jonathan that reviewed our English and polished our writing.

## Author Contributions

**Conceptualization:** Michelle Cristine Medeiros Jacob, Maria Fernanda Araújo de Medeiros, Ulysses Paulino Albuquerque.

**Data curation:** Michelle Cristine Medeiros Jacob, Maria Fernanda Araújo de Medeiros.

**Formal analysis:** Michelle Cristine Medeiros Jacob.

**Investigation:** Michelle Cristine Medeiros Jacob, Maria Fernanda Araújo de Medeiros.

**Methodology:** Michelle Cristine Medeiros Jacob, Maria Fernanda Araújo de Medeiros, Ulysses Paulino Albuquerque.

**Project administration:** Michelle Cristine Medeiros Jacob.

**Supervision:** Ulysses Paulino Albuquerque.

**Writing – original draft:** Michelle Cristine Medeiros Jacob, Maria Fernanda Araújo de Medeiros.

**Writing – review & editing:** Michelle Cristine Medeiros Jacob, Ulysses Paulino Albuquerque.

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
