## [Decision Letter · Decision Letter 0]

16 Apr 2020

Biodiverse food plants in the semiarid region of Brazil have unknown potential: A systematic review

PONE-D-20-06488

Dear Dr. Jacob,

We are pleased to inform you that your manuscript has been judged scientifically suitable for publication and will be formally accepted for publication once it complies with all outstanding technical requirements.

With kind regards,

Rainer W. Bussmann

Academic Editor

PLOS ONE

Additional Editor Comments (optional):

A truly excellent paper. All reviewers agreed that it essentially can be accepted as is. One suggested to include another reference, which this editor however deems unnecessary. As such, the paper is one of the few that can be accepted for publication right away.

Journal Requirements:

Please ensure that your manuscript meets PLOS ONE's style requirements, including those for file naming. The PLOS ONE style templates can be found at http://www.plosone.org/attachments/PLOSOne_formatting_sample_main_body.pdf and http://www.plosone.org/attachments/PLOSOne_formatting_sample_title_authors_affiliations.pdf

Reviewers' comments:

Reviewer's Responses to Questions

**Comments to the Author**

1. Is the manuscript technically sound, and do the data support the conclusions?

Reviewer #1: Yes

Reviewer #2: Yes

Reviewer #3: Yes

2. Has the statistical analysis been performed appropriately and rigorously? 

Reviewer #1: Yes

Reviewer #2: N/A

Reviewer #3: N/A

3. Have the authors made all data underlying the findings in their manuscript fully available?

Reviewer #1: Yes

Reviewer #2: Yes

Reviewer #3: Yes

4. Is the manuscript presented in an intelligible fashion and written in standard English?

Reviewer #1: Yes

Reviewer #2: Yes

Reviewer #3: Yes

5. Review Comments to the Author

Reviewer #1: The paper is written in good English and is great to read. It is a very interesting approach to wild foods. I enjoyed reading it.

I am not myself and expert on Brazilian flora so I cannot assess the sources used but from a perspective of someone studying wild foods the paper can be published nearly as it is. I would only add a few references on other paper discussing the potentially edible plants of an area, particularly:

Blanco-Salas, J., Gutiérrez-García, L., Labrador-Moreno, J. and Ruiz-Téllez, T., 2019. Wild plants potentially used in human food in the Protected Area" Sierra Grande de Hornachos" of Extremadura (Spain). Sustainability, 11(2), p.456.

Jug-Dujakovic M, Luczaj L. THE CONTRIBUTION OF JOSIP BAKICS RESEARCH TO THE STUDY OF WILD EDIBLE PLANTS OF THE ADRIATIC COAST: A MILITARY PROJECT WITH ETHNOBIOLOGICAL AND ANTHROPOLOGICAL IMPLICATIONS. Slovensky Narodopis. 2016;64(2):158.

Reviewer #2: The article is well written and can be of interest for many readers. Authors guide attention to the important shortcomings in data collection on wild food plants and although the results could cover wider scope of plants, the limitations of the study are well explained.

Reviewer #3: The manuscript has been very well developed, the objectives are quite clear and supported by a solid (and replicable) methodology have allowed us to present interesting and very solid results. The discussions provide an interesting perspective on the species that have been selected. It is highly appreciated that authors identify the limitations of their work. Congratulations to the authors for their solid and interesting work.

6. PLOS authors have the option to publish the peer review history of their article (what does this mean?). If published, this will include your full peer review and any attached files.

Reviewer #1: Yes: Łukasz Łuczaj

Reviewer #2: No

Reviewer #3: No

---

## [Editor Report · Acceptance letter]

17 Apr 2020

PONE-D-20-06488 

Biodiverse food plants in the semiarid region of Brazil have unknown potential: A systematic review 

Dear Dr. Jacob:

I am pleased to inform you that your manuscript has been deemed suitable for publication in PLOS ONE. Congratulations! Your manuscript is now with our production department. 

With kind regards,

on behalf of

Dr. Rainer W. Bussmann 

Academic Editor

PLOS ONE